# Cost-Effectiveness Analysis of Universal Rotavirus Vaccination Schedules in Syria

**DOI:** 10.3390/vaccines13111157

**Published:** 2025-11-12

**Authors:** Mania Mershed, Razan Altarabishi, Rasha Mohamed, Lamia Abu ajaj, Dima Alrashee, Manar Kamel, Salah Al Awaidy

**Affiliations:** 1Department of Research, MOH, Damascus, Syria; maniamershed2020@gmail.com (M.M.); rashamohamed2008@gmail.com (R.M.); 2Department of Primary Health Care, MOH, Damascus, Syria; razan.altarabishi@gmail.com (R.A.); lamia.ahmad.abu.ajaj@gmail.com (L.A.a.); 3Department of Health Economy, MOH, Damascus, Syria; dimaincidental@gmail.com (D.A.); dr.manarkamel@yahoo.com (M.K.); 4Freelance Public Health Consultant, Muscat 111, Oman

**Keywords:** rotavirus, rotavirus vaccine, cost-effectiveness, gastroenteritis, quality-adjusted life years, DALY, national immunization program, Syria

## Abstract

**Highlights:**

**Abstract:**

**Background:** Rotavirus (RV) continues to be the leading cause of acute gastroenteritis (AGE) globally among children under five. National RV vaccination efforts have lowered morbidity and mortality. Vaccination is a key public health tool to alleviate this substantial burden of RV in middle- and low-income countries. In Syria, RV morbidity accounts for 27% of severe GE. We conducted a cost-effectiveness analysis of introducing rotavirus vaccinations (RVV) into Syria’s National Immunization Program. **Methods:** A decision tree model was developed to assess the cost-saving of two-dose rotavirus vaccinations (Rotarix^®^) compared to no vaccination. A birth cohort of 573,944 newborns was simulated throughout a 5-year time frame to capture the near-term health and economic effects. The analysis adopted an incremental cost-saving approach, evaluating a hypothetical 2023 birth cohort from the government’s perspective. Outcomes included the cost per disability-adjusted life year (DALY) prevented and the cost per death averted. Model inputs were derived from local data, specifically including healthcare and vaccination costs and deaths attributable to RVGE, the scientific literature, and national/international databases. The incremental cost-effectiveness ratio (ICER) measures the cost of avoiding one disability-adjusted life year (DALY) adopted. **Results:** Over five years, the two-dose RV strategy would avert 77,500 RVGE cases, reduce outpatient visits by 59%, and reduce severe RV hospitalizations by 41%. The vaccination program would cost $21,817,918 USD and avert $3,239,907 USD in healthcare costs, resulting in a net cost of $18,578,011 USD. The incremental cost-effectiveness ratio (ICER) was $2098 USD per DALY averted, which is below three times Syria’s GDP per capita ($753.6 USD), indicating high cost-effectiveness according to WHO benchmarks. **Conclusions:** Introducing rotavirus vaccination is highly cost-saving and will result in a substantial reduction in healthcare burdens and lives lost. Policy planners must ensure its inclusion in the National Immunization Programs, ensuring sustainable financing and equitable access.

## 1. Introduction

An estimated 1.17 million (95% UI 0.793–1.62) deaths across all age groups globally were attributed to diarrheal diseases in 2021 [1]. Although treatable and preventable, an estimated 30% of the global diarrheal mortality occurred in children under 5 years of age, with Sub-Saharan Africa accounting for 70% of the under-five global diarrheal mortality [1].

Rotavirus (RV) was the leading cause of diarrheal deaths in the Global Burden of Disease (GBD) 2019 study, responsible for 19.11% (*n* = 235,331 ranging from 110,221 to 415,457) deaths of diarrheal-related fatalities in 2019 [2]. Over the past three decades, rotavirus-related deaths have declined significantly, dropping from 659,053 (95% UI 314,974–1,125,598) in 1990 to 235,331 (95% UI 110,221–415,457) in 2019 [2]. Children with rotavirus gastroenteritis (RVGE) were more likely to experience symptoms such as dehydration, metabolic acidosis, and fever and have a higher likelihood of requiring hospitalization [3]. Children with rotavirus infection may also develop afebrile and febrile seizures, as well as transiently reduced consciousness, like those in children with non-rotavirus-associated gastroenteritis [3]. However, children with rotavirus gastroenteritis were more likely to have encephalopathy [3].

An estimated 24 million outpatient visits and 2.4 million hospitalizations each year have been attributed to RV infection in children [4]. In the United States, the estimated annual social and medical costs associated with rotavirus infections exceeded US $1 billion and US $264 million, respectively, whilst in Taiwan, the estimated yearly social and hospital costs for rotavirus-related hospitalizations were US $13.3 million and US $10.4 million [5]. In China, the social yearly costs for rotavirus-related hospitalizations were estimated at $76.05 billion, of which $55.64 billion was due to private hospital expenditure [6].

Several live-attenuated RV vaccines have been recommended to alleviate the burden of RVGE [7]. The World Health Organization (WHO) advocates for the incorporation of RV vaccinations into all national immunization programs (NIPs) and advises that the first dose of the RV vaccine be delivered promptly after six weeks of age, in conjunction with the diphtheria, tetanus, and pertussis (DTP) vaccination [8].

Incorporating rotavirus vaccinations into national immunization programs globally has resulted in a 59% reduction in rotavirus hospitalizations and a 36% decline in diarrhea fatalities attributable to rotavirus in countries that have used the vaccine. Furthermore, with the introduction of the RV vaccine, diarrhea fatality rates have dropped from 20.8% in 1990 to 12.6% in 2008, with a further drop to 10% in 2018. Similar trends have been observed in numerous middle- and low-income countries [9,10]. The WHO recommends integrating rotavirus vaccination into national Expanded Programmes on Immunization (EPIs), especially where diarrhea is a primary cause of child mortality [11].

WHO also emphasizes the importance of aligning vaccination programs with national health goals, involving political and community stakeholders, and implementing surveillance systems to monitor vaccine coverage, safety, and cost-effectiveness (CE). By adhering to these guidelines, countries can significantly reduce rotavirus-associated illness and mortality while improving public health outcomes [11]. Multiple countries have conducted CE studies on rotavirus immunization [9,10,12]. Nevertheless, its cost-effectiveness has never been established in Syria.

Conflict has affected vaccination in Syria and vaccination rates dropped to almost half by 2013 in some areas due to the ongoing war, especially for diseases like diphtheria, tetanus, and pertussis, as they are considered to be the main vaccine-preventable diseases [13]. During the period 2021–2023, the national surveillance of gastroenteritis sentinel sites indicated 784 severe gastroenteritis hospitalizations over the 3 years among children less than 5 years of age, and rotavirus gastroenteritis (RVGE) was responsible for 27% of diarrheal hospitalizations in this age group and imposes a substantial health and economic burden on Syria, There is no reported mortality associated with rotavirus (reported by the rotavirus program, MOH). The lack of mortality reports could be ascribed to underreporting or insufficient monitoring systems in conflict settings throughout the country.

Country-specific cost-effectiveness analyses are essential for evidence-based policymaking, especially in resource-constrained settings. This study aims to fill this critical evidence gap by evaluating the cost-effectiveness of introducing a universal, two-dose rotavirus (Rotarix^®^) vaccination schedule into Syria’s National Immunization Program. Using a decision-tree model and incorporating the best available local data, this analysis seeks to inform national health authorities and international partners on the potential health benefits, economic impact, and strategic value of a rotavirus vaccination program in Syria.

## 2. Methods

### 2.1. Model Overview

This study employed a decision tree model to estimate the costs and health effects of a rotavirus (RV) vaccination program over five years, applied to a hypothetical annual birth cohort of children. A decision-tree analytic model was developed to evaluate the cost-saving of introducing a universal, two-dose rotavirus vaccination schedule (Rotarix^®^) into Syria’s National Immunization Program (NIP), compared to the current practice of no vaccination. The vaccine was assumed to be administered to 85% of a hypothetical annual birth cohort of 573,944 children, resulting in 487,853 vaccinated children. All costs and health outcomes were discounted at an annual rate of 3%.

The analysis was conducted from the government healthcare system perspective, with a supplementary analysis from the societal perspective included in the sensitivity analysis. The model evaluated the cost-effectiveness (CE) and benefit of implementing a routine infant rotavirus vaccination program compared to the absence of vaccination, using data collected in 2023. The analysis focused on the nationwide introduction of a mandatory two-dose rotavirus vaccine, Rotarix^®^ (manufactured by GlaxoSmithKline, Rixensart, Belgium), administered at 3 and 5 months of age. The model projected outcomes such as RV cases, medical visits, hospital admissions, fatalities, immunization program costs, and both societal (applied for home-management) and healthcare system expenses.

A 3% annual discount rate was applied to both costs and benefits to account for the time value of money and health impacts. All monetary estimates were adjusted to 2023 USD. Inputs for the model were derived from a combination of country-specific data and extrapolated estimates from similar settings in other countries. The analysis was conducted from the perspectives of both government healthcare expenditures and societal costs (applied for home-managed). The financial implications of RV vaccination were assessed by contrasting the costs associated with implementing the vaccination program against the savings achieved through reductions in the RVGE burden. The primary outcomes included the number of fatalities prevented and disability-adjusted life years (DALYs) averted through vaccination.

Estimates were based on both country-specific data and extrapolation of missing data from the data of other countries. The impact of RV vaccination on financial implications was divided into costs related to the implementation of the vaccine program and savings achieved due to the reduction in the RVGE burden. The main output of the analysis encompasses prevented fatalities, averted DALYs, RV cases, hospital and outpatient visits, informal “visits”, and societal and government costs attributable to the RV vaccine. Supplementary outputs included the overall expenditure of the immunization campaign and the cost per DALYs avoided.

### 2.2. Model Structure (Decision Tree Model)

The model structure, depicted in Figure 1, compares two mutually exclusive strategies:

Vaccination Strategy: Infants receive two doses of Rotarix^®^ at 3 and 5 months of age, concomitant with other routine vaccinations.

No-Vaccination Strategy: The status quo, with no rotavirus vaccination program.

For both groups, the cohort assesses a probability of developing rotavirus gastroenteritis (RVGE), its severity (non-severe or severe), pathway of care (outpatient visit, hospitalization, or home care) and mortality. The model explicitly calculates the associated costs and disability-adjusted life years (DALYs) for each pathway in both arms, allowing for a direct comparison of the difference, if any.

### 2.3. Model Inputs

Model parameters were derived from a comprehensive review of the scientific literature, international databases, and, crucially, nation-specific data provided by the Syrian Ministry of Health (MOH). The key input parameters are summarized in Table 1.


**Demographic and disease burden assumptions data**


The annual birth cohort was estimated at 573,944, based on MOH projections for 2023. Vaccine coverage for the two-dose series was assumed to be 85%, aligned with the WHO/UNICEF reported 2022 coverage for the second dose of pentavalent vaccine in Syria. The incidence of RVGE was based on sentinel surveillance data from five major governorates (Damascus, Lattakia, Deir ez-Zor, Hama, and Aleppo), selected because they house the most prominent hospitals and represent areas where children under five make up approximately 70% of the population.

An annual incidence of 27,000 per 100,000 children (equating to a 27% probability) under five was observed. The distribution of cases by severity (69% non-severe, 31% severe) and the probabilities of seeking formal care were sourced from the published literature [14,15]. The hospitalization rate for severe RVGE was estimated at 23.8 per 100,000 children [16].

Disease burden estimates were derived from these governorates, with the Ministry of Health (MOH) providing specific data on the duration and costs of treatment for RVGE for the last three years.

The average duration of hospitalization for RVGE was estimated at 4 days, with an associated cost of USD 343 per hospitalization, discounted at 3% over five years. Outpatient visits were estimated to cost USD 6 per visit under similar discounting assumptions. For children managed with home care, it was assumed that they incurred no direct healthcare costs. The MOH estimated the RV incidence among children under the age of five to be 27,000 per 100,000 (Range 24,000–28,000), as well as RVGE age distributions of cases.

Severity is determined by the length and frequency of loose stools, as well as the duration of vomiting, dehydration, and fever [17]. Walker CLF et al. estimated that non-severe cases RVGE account for 69% of all diarrhea episodes, while severe cases account for 31% [14]. Debellut F et al. assessed outpatient (clinic) visits for non-severe RVGE patients at 10%, while severe cases were assessed at 100% [15]. Furthermore, it was estimated that the average hospitalization rate due to rotavirus is 23.8 (interquartile range: 21.0–65.9) per 10,000 children under five years old in countries with an average mortality rate [18]. Our model did not account for re-infection to preserve a conservative estimate of the cumulative incidence. The case-fatality rate for children with RVGE was calculated based on the predicted global mortality from severe RVGE, which is 33 per 100,000 [19] and was estimated to be 33 deaths nationally (Table 1).

The primary outcomes analyzed included the number of fatalities prevented, DALYs averted, RVGE cases avoided, reductions in hospitalizations and outpatient visits, and informal caregiver visits. Additionally, the study assessed societal (applied for home-management) and governmental cost savings attributable to the vaccination program. Supplementary outcomes included the total costs of the immunization program and the cost per DALYs averted. These results provided a comprehensive evaluation of the vaccination program’s impact on health and economic outcomes.

**Table 1 vaccines-13-01157-t001:** The key model input parameters for rotavirus gastroenteritis burden.

Parameter	Central Value	Scenarios		Source
		Lower Input	Higher Input	
Incidence (per 100,000 under-five children)				
Overall incidence of RVGE	27,000	23,000	30,000	MOH
Non-severe RVGE cases (69%)	18,630	15,543	20,521	[14]
Non-severe RVGE institutional visits				
(10% of the 69%)	1863	15,783	20,210	[15]
Severe RVGE cases	8370	6678	10,862	[14]
Severe RVGE institutional visits (100%)	8370	6678	10,862	[15]
Severe RVGE hospitalizations	238	193	321	[16]
Severe RVGE deaths (33/100,000)	33	3	33	[19]
Disability weights				
Non-severe RVGE	0.18	0.12	0.26	[4]
Mean duration of illness in days for non-severe RVGE	4	3	5	MOH
Severe RVGE	0.24	0.13	0.34	[4]
Mean duration of illness in days for severe RVGE	7	6	8	MOH
RVGE age distribution	Cumulative percentage (%)			
<1 month	0			MOH
<2 months	7			MOH
<3 months	13			MOH
<6 months	36			MOH
<1 year	71			MOH
<2 years	94			MOH
<3 years	97			MOH
<4 years	99			MOH
<5 years	100			MOH

RVGE, Rotavirus gastroenteritis. MOH, Ministry of Health.

The model assumed a two-dose RV vaccine schedule (Rotarix^®^) administered at 3 and 5 months of age to a hypothetical birth cohort of 487,853 out of 573,944 children (this is the cohort that could be reached at this time due to the ongoing conflict in the country and based on DTP3). Vaccine coverage was estimated at 85% based on 2022 coverage rates for Syria’s second dose of pentavalent and MMR vaccines. Vaccine efficacy was derived from clinical trial data, assuming a two-dose course provides 58% protection against severe rotavirus gastroenteritis [18]. Current coverage of the second dose of pentavalent at 3 months and measles, mumps, and rubella (MMR) at 5 months at 85% was used to forecast the anticipated RV vaccine coverage within a national vaccination initiative.

### 2.4. Vaccine Efficacy and Health Outcomes

While clinical trial efficacy was high, we used a lower real-world effectiveness estimate from similar settings to ensure conservatism; the vaccine efficacy against severe RVGE was set at 58%, based on real-world effectiveness data from comparable low-income country (LIC) settings [20]. This efficacy was applied to reduce the probability of developing severe RVGE in the vaccinated arm of the model.

The primary health outcome was the number of disability-adjusted life years (DALYs) averted. DALY weights of 0.18 and 0.24 were applied for non-severe and severe RVGE episodes, respectively, with durations of 4 and 7 days [4,14]. Given the low number of reported rotavirus-attributable deaths in Syria, the base-case analysis conservatively assumed a mortality of 33; the impact of potential mortality was tested in sensitivity analysis.

### 2.5. Vaccine-Related Costs and Assumptions

Costs were evaluated from both the healthcare system and some societal perspectives, and all costs are reported in 2023 U.S. dollars (USD).

Vaccination program costs: These included the vaccine price per dose ($2.12, UNICEF 2024 price [21]), a 5% wastage factor, and incremental system costs for training personnel, social mobilization, cold chain expansion, and other operational expenses as estimated by the Syrian MOH (Table 2).Healthcare Costs: The MOH provided the average cost per RVGE hospitalization ($343) (the cost including physician examinations, testing expenses, admission, and medication) and per outpatient visit ($6) (the cost including physician examinations, testing expenses, and medication) over the 5-year duration. For the societal perspective analysis, a conservative cost of $2.50 was applied for home-managed cases to account for oral rehydration salts and antipyretics.

The costs of RV vaccines are presumed to be incurred during the first year of life for the birth cohort, and the costs are not influenced by discounting. The full per-course price of the vaccines (USD 2.12 for Rotarix^®^), as per the UNICEF price and the model’s base-case scenario, assumes a stable value over the period 2023 to 2030. The vaccine is in a one-dose vial, in a liquid presentation. After carefully considering the cost per dosage, waste, and volume, the cost of introducing the rotavirus vaccine was determined by the number of target populations, the number of doses required, and the cost of the vaccines. Additional costs, such as transportation, supply chain, and related training, were also calculated.

The total number of vaccines was calculated as total target population (573,944) × 2 doses of vaccines × 1.05 (wastage factor as per WHO) × coverage of 2 doses (85%) = 1,024,490 doses. The estimation of rotavirus cost based on the 2024 UNICEF estimation is 2.12 US$ [21]. Hence, the cost of the rotavirus vaccine = 10,280,613 × 2.12 = 21,714,918. Program costs: US$ 103,000, covering personnel, training, cold chain equipment, and other operational expenses [based on rotavirus program, MOH estimation]. The estimated cost of the introduction of the vaccine is 21,817,918 US$ (Table 2).

Personnel, maintenance and overhead, short-term training, information, education, and communication (IEC) and social mobilization, disease monitoring, program administration, other regular recurring expenses, vehicles, and other new cold chain equipment are among the spending categories that are pertinent to rotavirus and were estimated by the Syrian NIP, MOH.

### 2.6. Disease-Related Costs and Assumptions

We differentiate cases into those seeking formal or informal care. The MOH estimated a comprehensive breakdown of the costs that households and health services incur as a result of RVGE cases. Formal outpatient or formal inpatient care will result in costs for providers (e.g., the government).

Health service costs of RVGE among children < 5 years (cost per child) were estimated by the MOH. The household average cost of non-severe RVGE rotavirus cases visiting the facility (outpatient), the average cost of severe RVGE cases visiting the facility (outpatient), and the average cost of hospitalization for severe RVGE cases were estimated. For non-severe and severe RVGE rotavirus, household expenses related to all social perspectives were estimated to be 6 and 12 USD, respectively. The estimated average cost of hospitalization for severe RVGE cases and the cost of health services was USD 343.

Analytical Methods: The cost-effectiveness and cost–benefit analysis was determined as follows:


**Cost-Effectiveness Analysis**


The primary outcome was the incremental cost-effectiveness ratio (ICER), calculated as follows:

ICER = (Cost_Vaccination − Cost_NoVaccination)/(DALYs_Vaccination − DALYs_NoVaccination)

The intervention was considered highly cost-effective if the ICER was less than Syria’s GDP per capita ($753.6 USD) and cost-effective if less than three times the GDP per capita, as per WHO and Gavi guidelines [22].

### 2.7. Cost–Benefit Analysis

A secondary benefit–cost ratio (*BCR*) was calculated to summarize the economic return:

*BCR* = (Costs_Averted_by_Vaccination)/(Costs_of_Vaccination_Program)

A *BCR* > 1 indicates that the economic benefits outweigh the program costs.

### 2.8. Model Outputs


**Effects**


The assessment of the economic viability of the Rotarix^®^ vaccination in the research was performed using two methodologies:

The first method was the benefit–cost ratio (*BCR*) analysis to describe the overall connection between the relative costs and benefits of the proposed intervention. The first approach is the benefit–cost ratio (*BCR*) analysis, which was employed based on the below equation.(1)$$BCR=\sum_{i=1}^{n}\frac{B}{(1+r{)}^{i}}/C$$

*B* = benefit or cost in year, *n* = number of years in the evaluation period, *r* = real discount rate, *C* = the cost of intervention.

The second method used is the health economic assessment of the Rotarix^®^ vaccination using cost-effectiveness analysis. The cost-effectiveness of delivering two doses of rotavirus vaccine as part of the national immunization program for children under the age of five was estimated. The clinical and economic results of the intervention (vaccine administration) were compared to those of non-vaccination. In this strategy, the major outcome measure is the incremental cost-effectiveness ratio (ICER), which measures the cost of avoiding one disability-adjusted life year (DALY) [23].


**Sensitivity analysis**


We performed sensitivity analyses to determine how parameter uncertainty affects the base-case incremental cost-effectiveness ratio (ICER), DALYs, and costs. Using lower and higher estimates of each parameter, univariate deterministic sensitivity studies reveal the most sensitive parameters of the ICER. Due to joint uncertainty in modeled parameters, probabilistic sensitivity analyses (PSAs) assessed the uncertainty. The sampled values were used to compute additional costs, DALYs, and ICER 1000 times to provide uncertainty distributions for deterministic outcomes. A cost-effectiveness plane depicts the simulated combined uncertainty of incremental costs and DALYs (Table 2). To account for variability, we now use a plausible range of 50% to 82% in sensitivity analyses, informed by a recent meta-analysis of rotavirus vaccine performance in LMICs [15].


**Ethical approval**


The study was approved by the Syrian Ministry of Health Research Committee (Approval number 1690). The analysis utilized aggregated, anonymized data available in the public domain, complying with the Declaration of Helsinki.

## 3. Results

### 3.1. Base-Case Analysis: Health Outcomes

Over the five-year time horizon, the introduction of rotavirus vaccination for a single birth cohort would avert a substantial number of RVGE cases and related healthcare events, compared to the no-vaccination strategy (Table 3). Thus, an estimated 77,500 RVGE cases would be averted, outpatient visits would be reduced by 59%, and severe RV hospitalizations would be reduced by 41%. The vaccination program would cost $21,817,918 USD and avert $3,239,907 USD in healthcare costs, resulting in a net cost of $18,578,011 USD. The incremental cost-effectiveness ratio (ICER) was $2098 USD per DALY averted, which is below three times Syria’s GDP per capita ($753.6 USD), indicating high cost-effectiveness according to WHO benchmarks. Meanwhile the model conservatively assumed a base-case scenario of 33 rotavirus-attributable mortalities among severe cases, which implies a likely reduction in mortality risk.

The total cost of implementing the vaccination program, including vaccine procurement and delivery, was estimated at $21,817,918 USD. This investment resulted in savings of $3,239,907 USD in direct medical costs, primarily from averted hospitalizations and outpatient visits. From a cost–benefit perspective, this yielded a benefit–cost ratio (BCR) of 0.15, indicating that the direct medical cost savings cover 15% of the vaccination program’s cost.

The primary cost-effectiveness analysis, from the government perspective, showed that the rotavirus vaccination strategy averted 300 DALYs at a net cost of $18,578,011 USD. The resulting incremental cost-effectiveness ratio (ICER) was $2098 per DALY averted. This is below three times Syria’s GDP per capita ($2261), qualifying the intervention as highly cost-effective according to WHO benchmarks.

### 3.2. Base-Case Analysis: Economic Outcomes and Cost-Effectiveness

The total cost of implementing the vaccination program, including all logistics associated with vaccine procurement and delivery, was estimated at 21.86 million USD. This investment, however, resulted in savings in ample healthcare costs. The reduction in RVGE cases averted an estimated 3.24 million USD in direct medical costs, mainly from averting hospitalizations and avoiding outpatient visits.

The primary cost-effectiveness analysis, from the government perspective, showed that rotavirus vaccination was cost-saving. The vaccination strategy was both more effective (averting 444 DALYs) and economical than the no-vaccination strategy, resulting in a negative incremental cost-effectiveness ratio (ICER). The net cost savings were $373,389 USD, with an ICER of −$840 per DALY averted.

### 3.3. Sensitivity Analysis


**Deterministic Sensitivity Analysis (DSA)**


The model was most sensitive to the price of the vaccine per dose and the incidence rate of RVGE. Varying the vaccine price between USD 1.59 and USD 2.65 resulted in an ICER range of $1550 to $2750 per DALY averted. Varying the incidence rate resulted in an ICER range of $1800 to $2450. In all scenarios, the ICER remained below the $2261 threshold for high cost-effectiveness.


**Probabilistic Sensitivity Analysis (PSA)**


The PSA confirmed the robustness of the base-case finding. At a willingness-to-pay threshold equal to Syria’s GDP per capita ($753.6), the probability of vaccination being cost-effective was 12%. However, at the more relevant threshold of three times GDP per capita ($2261), the probability of cost-effectiveness was 92%. When including patient transportation and caregiver costs from the societal perspective, the ICER became more favorable ($1850 per DALY averted). Applying a conservative mortality rate (33 deaths/100,000) also improved the ICER to $1450 per DALY averted, with 15 deaths averted over the 5-year period.

The model was most sensitive to the price of the vaccine per dose and the incidence rate of RVGE. Varying the vaccine price between USD 1.59 and USD 2.65 resulted in an ICER range of 550 to 1130 USD per DALY averted. Similarly, varying the incidence rate resulted in an ICER range of USD 620 to USD 1150. Despite this variation, the ICER remained below the WHO cost-effectiveness threshold (the WHO-recommended threshold of 1–3 times GDP per capita per DALY averted) of USD 753.6 per DALY averted (Syria’s GDP per capita) in all scenarios for these key parameters, confirming the robustness of the base-case conclusion.

## 4. Discussion

Our cost-effectiveness model analysis demonstrates that the introduction of a universal two-dose rotavirus vaccination program would prove to be a highly cost-saving public health intervention in Syria. The program is projected to avert a significant portion of the RVGE burden, including 77,500 cases, 59% of outpatient visits, and 41% of hospitalizations and mortality over five years. With an incremental cost-effectiveness ratio (ICER) of USD 2098 per DALY averted, the intervention falls well below the threshold of three times Syria’s GDP per capita, qualifying as cost-effective according to WHO and Gavi guidelines [15,22]. While the benefit–cost ratio of 0.15 shows the investment is not cost-saving from a direct medical cost perspective, it represents a highly efficient investment in child health. These findings align consistently with economic evaluations of rotavirus vaccination in other LMICs facing similar epidemiological profiles. These findings align consistently with economic evaluations of rotavirus vaccination in other LMICs facing similar epidemiological profiles. The reported ICERs from countries like Pakistan (USD 165–200/DALY) [22,24], Kenya (USD 288/DALY) [22,25] and other Asian nations (USD 212–1653/DALY) [26] uphold that rotavirus vaccination is an economically sound investment in resource-constrained settings. Despite the ongoing humanitarian catastrophe, the vaccine’s favorable cost-effectiveness in Syria highlights how resilient vaccination programs are and how they may still have a major positive impact on health, even in weak health systems. The expanding body of research in favor of rotavirus vaccine introduction in the Eastern Mediterranean Region is strengthened by our analysis, which provides important, context-specific evidence.

The reliance of our study on nationally generated data for important metrics, such as RVGE incidence, healthcare consumption, treatment costs, and vaccine coverage forecasts, is one of its main strengths. This ensures that the model is based on the real-world local health system realities in Syria. In addition, we used a transparent decision-tree model and carried out thorough sensitivity studies, which validated the validity of our main finding, that of rotavirus vaccination being cost-effective across a variety of assumptions.

Despite these notable strengths, our study has several limitations. First of all, our conservative model did not factor in herd immunity effects, which are well-documented for rotavirus vaccines [1]. Inclusion of these indirect benefits would likely improve the ICER further. This would suggest that our estimates possibly understate the absolute effects of rotavirus vaccination. Secondly, our analysis primarily adopted a government perspective. The exclusion of indirect societal costs, such as lost productivity of the care-giver, and out-of-pocket expenses, omits a substantial portion of the economic burden of the rotavirus infection and thus underestimates the full economic benefit of the vaccination. Also, the static cohort model deployed by us assumes constant vaccine efficacy over time, but this might not be the case. Future research must employ dynamic transmission models to capture varying disease incidence, herd effects, and the long-term impact of vaccination, including those involving potential strain replacement or a waning immunity. The turbulent economic situation in Syria also introduces uncertainty in cost projections. As such, continuous monitoring and real-world evidence studies following the introduction of vaccine shall be crucial to validate our model-based projections and help refine the economic evaluation. Further, we have used WHO-CHOICE thresholds in our paper, in accordance with their widespread use in the literature. These thresholds have been widely criticized for being arbitrary, not reflecting opportunity costs, and risking misallocation of scarce resources, especially in low- and middle-income countries [26,27,28,29,30]. Alternative approaches, such as those proposed by Claxton et al. (2015) [29] and Woods et al. (2016) [30], attempt to derive context-specific cost-effectiveness thresholds from empirical estimates of health system marginal productivity. Furthermore, recent evidence suggests that even in high-income settings there is no consensus on standard thresholds. For example, Tzanetakos et al. (2023) [31] found that most economic evaluations in Greece continued to adopt the 1–3 × GDP per capita rule, despite the absence of methodological justification. This underlines both the persistence of the WHO thresholds in practice and the pressing need for more locally appropriate, evidence-based willingness-to-pay thresholds. Other study limitations are that the analysis was supplemented with a study of social factors applicable only to home-management. Incorporating wider societal benefits, such as caregiver time, saved productivity losses, long-term sequelae, and outbreak-control savings, would augment the numerator and potentially significantly elevate the benefit–cost ratio (BCR).

Thus, the introduction of rotavirus vaccination in Syria represents a calculated opportunity to mitigate mortality and morbidity in children and also assuage pressure on a conflict-ridden splintered healthcare system. However, successful implementation of the program shall require overcoming significant logistical challenges, including a including maintaining a strong cold chain and ensuring equitable vaccine access. We strongly recommend that Syrian health authorities and important stake holders like the Gavi, The Vaccine Alliance, etc., utilize our findings to prioritize the inclusion of rotavirus vaccination in the National Immunization Program, securing investments for the long-term viability of program. Parallel strengthening of surveillance systems to monitor coverage, impact, and safety is also emphasized.

## 5. Conclusions

Our study provides a compelling evidence that incorporating rotavirus vaccination into Syria’s National Immunization Program would be a cost-saving intervention, reducing the economic and health burden of RVGE in children under five. However, successful implementation will require careful financial planning, robust supply chain management, and continuous program evaluation. Policymakers need to explore sustainable financing mechanisms, inflation, pricing strategies, national health priorities, optimize vaccine procurement strategies, and ensure equitable vaccine access to maximize the public health impact. Future research should focus on refining cost estimates, evaluating long-term program effectiveness, and incorporating a broader societal perspective to provide a more comprehensive assessment of the intervention’s value.

## Figures and Tables

**Figure 1 vaccines-13-01157-f001:**
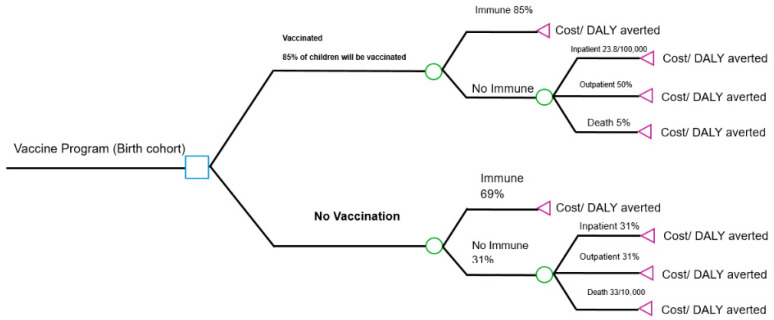
Decision tree model structure for the evaluation of rotavirus vaccination in Syria.

**Table 2 vaccines-13-01157-t002:** Input parameters for estimating health service costs of rotavirus gastroenteritis (RVGE) among children < 5 years (cost per child) for 5 years, presented in USD, 2023.

Parameter	Cost in USD	Source
Cost of rotavirus vaccine (two doses)	21,714,918	MOH
Training/supervision cost	61,000	MOH Estimation
Cost of information, education, and communication (IEC) and social mobilization, including developing training materials	27,000	MOH Estimation
Cost of installing cold chain equipment for rotavirus vaccine storage (based on UNICEF estimates), including safety boxes.	15,000	MOH Estimation
Total estimated cost	21,817,918	_

USD: US dollar.

**Table 3 vaccines-13-01157-t003:** Projected health outcomes and healthcare utilization averted over 5 years.

Outcome	No Vaccination Strategy	Vaccination Strategy	Events Averted	% Reduction
RVGE cases	310,000	232,500	77,500	25%
- Non-severe cases	213,900	160,425	53,475	25%
- Severe cases	96,100	72,075	24,025	25%
Outpatient Visits	22,455	9255	13,200	59%
Hospitalizations	1190	705	485	41%
DALYs lost	1200	900	300	25%

Note: Numbers are rounded for presentation. The 25% reduction in cases reflects the combined effect of 85% vaccine coverage and 58% vaccine efficacy.

## Data Availability

The data will be provided upon reasonable request.

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
