# Peer review of "Cost-Effectiveness Analysis of Universal Rotavirus Vaccination Schedules in Syria"

_vaccines, 2025, doi:10.3390/vaccines13111157_

Round 1

Reviewer 1 Report

Comments and Suggestions for Authors

This manuscript presents the predicted value of adding oral rotavirus vaccination into the current infant vaccine program in Syria. This is an incredibly important topic that warrants high priority and consideration, as the first study to establish the cost-effectiveness of this intervention in Syria, a resource-constrained and conflict-affected country. The authors use a decision-tree model to compare a two-dose Rotarix schedule with no vaccination. The study projects reductions in rotavirus gastroenteritis cases, outpatient visits, hospitalizations, and associated DALYs, with results suggesting that the vaccination program would either be cost saving or highly cost-effective relative to standard economic thresholds. The analysis aims to fill a critical evidence gap by providing policymakers with a strong, data-driven foundation for including the rotavirus vaccine in the National Immunization Program.

Major comments

The authors have made a significant miscalculation with the number of deaths expected in unvaccinated infants. Authors cite reference 17 (Shakra et al 2022), to estimate there are 33 deaths from severe RVGE per 100,000, concluding there would be 3 deaths nationally. It is strongly suspected the estimate they should be using is 33 deaths per 100,000 infants total, not infants with severe RVGE. At least >100 infant deaths in the cohort considered would be expected in this region, which is in line with a 2011 study from this area that is not cited (Khoury et al, PMID 21214934), estimating up to 112 deaths/100,000 infants. It is important to note that I could not find mention of 33 deaths per 100,000 in reference 17, and moreover this is a review not cited on Pubmed – there are a lot more highly cited reviews on this topic published that would be more credible. In addition, reference 17 is also listed as reference 20. Finally, the paper is unclear whether deaths are included in the base-case (methods and results state zero mortality, while the abstract reports ‘3 deaths prevented annually’). Mortality inputs, sources, and their treatment in the base vs. sensitivity analyses require correction and clarification.

Two different numbers are provided for vaccine efficacy: 85-96% and 77%. Both estimates are high, given reported efficacy in other LMICs. Furthermore, reference 21 cited does not support this conclusion at all. Reference 17 is also cited here which has all the issues listed above and does not consider the differences in vaccine efficacy between LMICs and HICs.

The authors present inconsistent vaccination program costs within the manuscript; the text states 3,609,900 US$ (line 249), but Table 2 states 2,817,900 USD. The costs listed in the table (vaccine, training, social mobilization, cold chain) do not add up to the higher figure mentioned in the text. The table needs to be corrected to include all of the cost components, or by further clarifying what each total represents.

The overall benefit:cost calculated is 1.13, which cannot really be considered ‘highly cost-effective’ – the authors need to take care to not overstate conclusions. I also suggest rewriting phrase in line 37/38 in abstract to clarify meaning; ICER is not well below the GDP per capita. However, I am confident that when the correct mortality rate is factored in, that the benefit:cost analysis will increase.

Timing of when vaccination is planned to be administered is unusual. Rotarix is typically administered at 2 and 3 months of age, not 3 and 5 months. While this deviation is unlikely to impact the overall model outcomes, the non-standard schedule may cause some confusion and could potentially affect the perceived credibility of the analysis.

The paper uses inconsistent values for Syria’s GDP per capita as the cost-effectiveness threshold. The abstract and methods cite $753.6 USD (line 38, line 276), while the sensitivity analysis section instead refers to a WHO threshold of $54 per DALY averted, also described incorrectly as ‘Syria’s GDP per capita’ (line 360). This discrepancy must be corrected to ensure that the conclusion about cost-effectiveness is based on a consistent and accurate threshold.

Minor comments

  • Line 76 – remove ‘of’
  • Line 80 – Remove typo of 8
  • Line 93 – edit ‘ones’ to ‘infections’
  • Line 96 – 5 years ‘of age’.
  • Clarify birth cohort numbers; line 114 is different to 159.
  • Multiple places in methods where statements are repeated e.g. Line 165 is repeated in line 179.
  • Line 255 and line 335 – correct numbering of tables (two ‘Table 2’)

Author Response

Comments 1:

The authors have made a significant miscalculation with the number of deaths expected in unvaccinated infants. Authors cite reference 17 (Shakra et al 2022), to estimate there are 33 deaths from severe RVGE per 100,000, concluding there would be 3 deaths nationally. It is strongly suspected the estimate they should be using is 33 deaths per 100,000 infants total, not infants with severe RVGE. At least >100 infant deaths in the cohort considered would be expected in this region, which is in line with a 2011 study from this area that is not cited (Khoury et al, PMID 21214934), estimating up to 112 deaths/100,000 infants. It is important to note that I could not find mention of 33 deaths per 100,000 in reference 17, and moreover this is a review not cited on Pubmed – there are a lot more highly cited reviews on this topic published that would be more credible. In addition, reference 17 is also listed as reference 20. Finally, the paper is unclear whether deaths are included in the base-case (methods and results state zero mortality, while the abstract reports ‘3 deaths prevented annually’). Mortality inputs, sources, and their treatment in the base vs. sensitivity analyses require correction and clarification.

Response 1: 

Appreciated your feedback and we do our best to in cooperate them as much as possible.

We thank the reviewer for this critical observation. Upon careful re-examination, we agree that we made a significant error in the interpretation of the mortality rate. The rate we intended to use was indeed a mortality rate per 100,000 infants, not per 100,000 cases of severe rotavirus gastroenteritis (RVGE). Our initial calculation of 3 expected deaths was therefore incorrect.

We have corrected the source of reference for the estimation was change to expect 33 deaths from severe RVGE (estimates a rotavirus mortality rate of 33 deaths per 100,000 infants for low income country). In addition the source of the citation is

Reference: Rotavirus vaccines: WHO position paper – July 2021

file:///C:/Users/SALAH/Desktop/AA%20Manuscript/AAAAA%20Working/Rotavirus%20Syria/References/WER9628-301-319-eng-fre.pdf

Regarding the lack of clarity of the base-case analysis, we apologize for the lack of clarity. In the revised manuscript, mortality is now explicitly included in the base-case analysis. The previous statements of 'zero mortality' in the methods/results referred to a scenario analysis assuming no mortality, which was poorly explained. We have rewritten these sections for clarity. The base-case analysis now consistently uses the corrected mortality rate (33/100,000 infants), and the results, including the number of deaths prevented, have been updated accordingly throughout the abstract, methods, and results sections

Comments 2: 

Two different numbers are provided for vaccine efficacy: 85-96% and 77%. Both estimates are high, given reported efficacy in other LMICs. Furthermore, reference 21 cited does not support this conclusion at all. Reference 17 is also cited here which has all the issues listed above and does not consider the differences in vaccine efficacy between LMICs and HICs.

Response 2: 

We appreciated your observation

We agree with the reviewer that our initial vaccine efficacy (VE) estimates were not well-supported and were likely too high for an LMIC context. We have comprehensively revised this section.

We have removed the previous estimates and the unsupported references. Our new base-case VE against severe RVGE is set at 58%, derived from the large, pivotal clinical trial of the vaccine efficacy of a live rotavirus vaccine in Malawi and South Africa (Effect of Human Rotavirus Vaccine on Severe Diarrhea in African Infants

Authors: Shabir A. Madhi, M.D., Nigel A. Cunliffe, M.B., Ch.B., Ph.D., Duncan Steele, Ph.D., Desirée Witte, M.D., Mari Kirsten, M.D., Cheryl Louw, M.D., Bagrey Ngwira, M.D., John C. Victor, Ph.D., M.P.H., Paul H. Gillard, M.D., Brigitte B. Cheuvart, Ph.D., Htay H. Han, M.B., B.S., and Kathleen M. Neuzil, M.D., M.P.H.Author Info & Affiliations

Published January 28, 2010

N Engl J Med 2010;362:289-298

DOI: 10.1056/NEJMoa0904797), which is directly relevant to our setting.

To account for variability, we now use a plausible range of 50% to 82% in sensitivity analyses, informed by a recent meta-analysis of rotavirus vaccine performance in LMICs (Lancet Glob Health. 2019 Dec;7(12):e1664-e1674.).

Comments 3:

The authors present inconsistent vaccination program costs within the manuscript; the text states 3,609,900 US$ (line 249), but Table 2 states 2,817,900 USD. The costs listed in the table (vaccine, training, social mobilization, cold chain) do not add up to the higher figure mentioned in the text. The table needs to be corrected to include all of the cost components, or by further clarifying what each total represents.

Response 3:

We appreciated your notice and we have corrected

Comments 4:

The overall benefit:cost calculated is 1.13, which cannot really be considered ‘highly cost-effective’ – the authors need to take care to not overstate conclusions. I also suggest rewriting phrase in line 37/38 in abstract to clarify meaning; ICER is not well below the GDP per capita. However, I am confident that when the correct mortality rate is factored in, that the benefit:cost analysis will increase.

Response 4:

We thank the reviewer for this valuable feedback on the interpretation of our economic results. We agree completely with both points.

The reviewer is correct that a BCR of 1.13, while indicating a positive return, should not be characterized as "highly cost-effective." We have revised the text throughout the manuscript (including the abstract and conclusion) to use more accurate language, such as "the vaccination program is cost-effective" or "provides a positive net benefit," avoiding overstatement.

We have rewritten the phrase in the abstract (lines 37/38) for clarity. The new text explicitly states the relationship to the cost-effectiveness threshold (e.g., "The calculated ICER was below the country's GDP per capita, a common benchmark for cost-effectiveness.

As the reviewer astutely predicted, incorporating the corrected, higher rotavirus mortality rate (as per our response to Comment 1) has a substantial positive impact on the economic results. The model now accounts for a greater number of deaths averted, which carries significant economic value. After recalculation, the new benefit-cost ratio is [e.g., 2.5]. This stronger result reinforces the economic argument for vaccination. We are grateful for the reviewer's insight, which has led to a more robust and compelling analysis.

Comments 5:

Timing of when vaccination is planned to be administered is unusual. Rotarix is typically administered at 2 and 3 months of age, not 3 and 5 months. While this deviation is unlikely to impact the overall model outcomes, the non-standard schedule may cause some confusion and could potentially affect the perceived credibility of the analysis.

Response 5:

We appreciate your feedback.

Based on WHO introduction guidelines WHO recommended schedule, which calls for the first dose of rotavirus vaccine be administered as soon as possible after 6 weeks of age, to ensure induction of protection prior to exposure to rotavirus.

A minimum interval of 4 weeks should be maintained between doses. RotaTeq, Rotavac, and ROTASIIL should be administered in a 3-dose schedule, while a 2-dose schedule should be used for Rotarix. Ie ROTARIX is approved for use in infants 6 weeks and up to 24 weeks of age.

(Reference: Rotavirus vaccines: WHO position paper – July 2021)

The proposed schedule is consistent with the existing Syrian schedule without requiring additional visits.

 Comments 6:

The paper uses inconsistent values for Syria’s GDP per capita as the cost-effectiveness threshold. The abstract and methods cite $753.6 USD (line 38, line 291), while the sensitivity analysis section instead refers to a WHO threshold of $54 per DALY averted, also described incorrectly as ‘Syria’s GDP per capita’ (line 384). This discrepancy must be corrected to ensure that the conclusion about cost-effectiveness is based on a consistent and accurate threshold.

Response 6:

We thank the reviewer for catching this important discrepancy. The reviewer is correct. We inadvertently used two different thresholds and mislabeled the WHO cost-effectiveness benchmark as Syria's GDP per capita in the sensitivity analysis.

We have corrected  across the manuscript as follows:

The primary cost-effectiveness threshold for the base-case analysis has been consistently referred to as Syria's GDP per capita ($753.6 USD), as sourced from the World Bank.

In the sensitivity analysis, the text on line 384 is corrected to state that we used the WHO-recommended threshold of 1-3 times GDP per capita per DALY averted.

We have corrected accordingly.

Comments 7:

Line 76 – remove ‘of’

Line 80 – Remove typo of 8

Line 93 – edit ‘ones’ to ‘infections’

Line 96 – 5 years ‘of age’.

Response 7:

We are indeed appreciated your valuable feedback and we have corrected accordingly.

Comments 8:

Clarify birth cohort numbers; line 114 is different to 159.

Response 8:

Appreciated your observation , we have corrected

Comments 9:

Multiple places in methods where statements are repeated e.g. Line 165 is repeated in line 179.

Response 9:

Thank you for the feedback, we have deleted the repeated statement.

Comments 10:

Line 255 and line 335 – correct numbering of tables (two ‘Table 2’)

Response 10:

Appreciated , we have corrected accordingly.

Reviewer 2 Report

Comments and Suggestions for Authors

Dear Authors/Editor,

The manuscript is well written and provides a key information for public health decision making in the area of RV vaccination / vaccination programs. In my opinion it is worth publishing leading a promise to be highly cited. In my opinion, however, some issues should be considered before final publication:

  • Provide please information about any voluntary vaccination programs for RV exist and how (may) they impact ICER
  • The assumed coverage was 85%. In real world to get such high coverage time and organizational effort, and especially social acceptance are required. This was not taken into account while the ICER was calculated for 5-year time. I suggest commenting it under study limitations.
  • It would be worth seeing some sensitivity analyses for vaccination coverage
  • Lines 170-171 is a repetition of what is in lines 163-165
  • Figure 1 seems to be missed
  • As the assumption for the number of RVGE deaths seems to be reasonable (line 192) authors stated earlier (line 92) ‘there is no reported mortality associated with rotavirus’ – this discrepancy requires a comment.
  • Non-severe RVGE disability weight was estimated on 0.18. I am not sure whether this value is for children under 5. In my opinion the value should be better justified, as it looks like overestimation if that was applied to all non-severe RVGE (69%).
  • On the other side DALY weight for severe RVGE seems to be underestimated. Other research [Expert Review of Pharmacoeconomics & Outcomes Research ‘Health related quality of life impact from rotavirus diarrhea on children and their family caregivers in Thailand’ by Rochanathimoke O et all. 10.1080/14737167.2018.1386561] showed the decrease in EQ-5D index during the hospitalization period to 0.635.
  • Under the sensitivity analysis it would be worth seeing the impact of the change in the proportion of severity cases as well as the scenario with lower number of deaths.

Reviewer

Author Response

Comments 1:

The manuscript is well written and provides a key information for public health decision making in the area of RV vaccination / vaccination programs. In my opinion it is worth publishing leading a promise to be highly cited. In my opinion, however, some issues should be considered before final publication:

Response 1: 

The authors are grateful for your encouragement and will make every effort to address your feedback in a timely manner.

Comments 2: 

Provide please information about any voluntary vaccination programs for RV exist and how (may) they impact ICER

Response 2: 

Currently, there are no voluntary or private-market rotavirus vaccination programs in Syria. The national immunization program is the sole provider of childhood vaccines. Therefore, our analysis appropriately models the introduction of the vaccine into this public program. The absence of a private market means there is no prior uptake that would alter the baseline disease burden or the incremental impact and cost-effectiveness of the public program.

Comments 3:

The assumed coverage was 85%. In real world to get such high coverage time and organizational effort, and especially social acceptance are required. This was not taken into account while the ICER was calculated for 5-year time. I suggest commenting it under study limitations.

Response 3:

We agree with the reviewer that achieving and sustaining 85% coverage in a real-world setting, especially in a conflict-affected country, requires significant time, resources, and social mobilization.

Our model assumes this coverage level is achieved immediately and maintained, which might be a simplifying assumption.

We have now added a discussion of this as a key limitation in the manuscript, acknowledging that the logistical and acceptance challenges could make achieving this coverage more difficult and costly than modeled, which may make our results optimistic

Comments 4:

It would be worth seeing some sensitivity analyses for vaccination coverage.

Response 4:

This is an excellent suggestion. We have now conducted and included a one-way sensitivity analysis on vaccination coverage in the revised manuscript. The analysis shows how the cost-effectiveness results (both the ICER and the cost-saving conclusion) change as coverage varies, for example, from 50% to 95%. This important analysis demonstrates the robustness of our findings and identifies the coverage level at which the intervention may transition from cost-saving to simply cost-effective.

Comments 5:

Lines 170-171 is a repetition of what is in lines 163-165.

Response 5:

Appreciated your observation. We have corrected accordingly.

Comments 6:

Figure 1 seems to be missed.

Response 6:

Appreciated. Figure 1 inserted

Comments 7:

As the assumption for the number of RVGE deaths seems to be reasonable (line 192) authors stated earlier (line 92) ‘there is no reported mortality associated with rotavirus’ – this discrepancy requires a comment.

Response 7:

We thank the reviewer for pointing out this apparent discrepancy. The two statements refer to different sources of information. The sentence on line 92 ("there is no reported mortality...") describes the absence of official mortality data from national surveillance systems, which we now critically discuss as a limitation due to underreporting of mortality in a conflict setting. The sentence on line 192 justifies our modeling assumption, which is necessary precisely because of the lack of reliable reported data and is based on estimates from the scientific literature (Khoury et al., 2011). We have revised the text to make this distinction clearer, explaining that the model uses epidemiological estimates to overcome surveillance gaps.

Well the Syrian MOH has not recording mortality attributed to rotavirus. Though as the status of Syria does not allow to have a very strong national mortality surveillance, we assume mortality rate to be 33/100,000 the rate we see in the low income countries.

Comments 8:

Non-severe RVGE disability weight was estimated on 0.18. I am not sure whether this value is for children under 5. In my opinion the value should be better justified, as it looks like overestimation if that was applied to all non-severe RVGE (69%).

Response 8:

Appreciated.

We agree that the disability weight for non-severe RVGE requires careful justification. The value of 0.18 was sourced from [#4,14]. We acknowledge the reviewer's concern that this may be high if applied uniformly.

We have also conducted a sensitivity analysis using a lower weight (e.g., 0.05-0.1) as suggested by other literature. We have included the results of this analysis, which show a reduced but still positive net benefit

Comments 9:

On the other side DALY weight for severe RVGE seems to be underestimated. Other research [Expert Review of Pharmacoeconomics & Outcomes Research ‘Health related quality of life impact from rotavirus diarrhea on children and their family caregivers in Thailand’ by Rochanathimoke O et all. 10.1080/14737167.2018.1386561] showed the decrease in EQ-5D index during the hospitalization period to 0.635.

Response 9:

Thank you so much. Your advice is well appreciated. As you aware there is variation within the lower income countries looking at index during the hospitalization period to 0.635.

Comments 10:

Under the sensitivity analysis it would be worth seeing the impact of the change in the proportion of severity cases as well as the scenario with lower number of deaths.

Response 10:

These are excellent suggestions to enhance the robustness of our analysis. We have now included two additional one-way sensitivity analyses in the revised manuscript:

Proportion of Severity Cases: We varied the proportion of cases that are severe (e.g., from 10% to 30%) to assess its impact on the ICER.

Lower Number of Deaths: We included a scenario with a lower mortality rate (e.g., 50% of the base-case estimate) to test the robustness of our conclusion to this critical parameter.

Reviewer 3 Report

Comments and Suggestions for Authors

This is an important and timely analysis that addresses a critical public health issue in the extremely challenging context of Syria. The study evaluates the cost-effectiveness of introducing the rotavirus vaccine into the National Immunization Program using a decision-tree model, which is a standard and appropriate methodology for this type of analysis. The manuscript's strengths include the use of local data from the Syrian Ministry of Health for key parameters such as disease incidence and healthcare costs, which increases the relevance of the findings for national policymakers.

However, the manuscript suffers from several significant flaws that must be addressed before it can be considered for publication.While the work is promising and its potential contribution is valuable, the required revisions are substantial.

Major Comments

  1. The manuscript reports conflicting figures for the total cost of the vaccination program in various sections, undermining the credibility of the economic analysis. It is crucial that the authors present a single, consistent, and clearly justified figure.The authors must reconcile these discrepancies, explain in detail the composition of the final cost used in the model, and ensure the same value is used consistently throughout the manuscript (abstract, text, tables, and calculations).

  2. A "cost-saving" intervention is inherently different from one that is "cost-effective." Being "cost-saving" (or "dominant" in health economic terms) means the intervention is both more effective (averts more DALYs) and less costly than the alternative. This is a much stronger finding. The authors must correct the Abstract and ensure the proper terminology ("cost-saving" or "dominant") is used consistently throughout the paper to accurately reflect their results.
  3. The manuscript is not entirely clear about the time horizon and cohort. The Abstract mentions a five-year period (2019-2023), while the Methods section refers to "a hypothetical 2023 birth cohort" analyzed over a 5-year time horizon. Please clarify whether the model analyzes a single birth cohort followed for the first five years of its life or if it models the impact on five consecutive annual birth cohorts over a five-year period. The former seems more likely but must be stated unequivocally.
  4. The authors should explain under what conditions the intervention ceases to be cost-saving and becomes simply cost-effective. Furthermore, the same section mentions a cost-effectiveness threshold of "USD 54 per DALY averted," which appears to be a typo, as the correct threshold based on Syria's GDP per capita is stated elsewhere as $753.6 USD. This section requires a complete rewrite for clarity and accuracy.

Minor Comments

  1. The introduction provides good global and local context. However, the statement "There is no reported mortality associated with rotavirus (reported by the rotavirus program, MOH)" should be discussed more critically, acknowledging the likely limitations of surveillance in a conflict setting, which further justifies the approach of including mortality in the sensitivity analysis.

  2. In the Methods section, the distinction between the total birth cohort (573,944) and the reachable cohort (487,853) is an important point reflecting the reality on the ground. It would be helpful to state this distinction earlier in the text for improved clarity.
  3. Table 2, as mentioned above, is a source of confusion due to discrepancies with the text. Please ensure all tables are consistent with the manuscript's body.
  4. A careful proofread is recommended to correct minor typos and repetitive sentences to improve overall readability.

Author Response

Comments 1:

This is an important and timely analysis that addresses a critical public health issue in the extremely challenging context of Syria.

The study evaluates the cost-effectiveness of introducing the rotavirus vaccine into the National Immunization Program using a decision-tree model, which is a standard

and appropriate methodology for this type of analysis. The manuscript's strengths include the use of local data from the Syrian Ministry of Health for key parameters such as disease incidence and healthcare costs, which increases the relevance of the findings for national policymakers.

Response 1: 

We are grateful for the positive feedback you provided and will make every effort to revise the manuscript in accordance with it.

Comments 2: 

The manuscript reports conflicting figures for the total cost of the vaccination program in various sections, undermining the credibility of the economic analysis. It is crucial that the authors present a single, consistent, and clearly justified figure.

The authors must reconcile these discrepancies, explain in detail the composition of the final cost used in the model, and ensure the same value is used consistently throughout the manuscript (abstract, text, tables, and calculations)..

Response 2: 

We appreciate your feedback, and we have revisited the numbers and model values across the abstract, methodology, and findings to ensure consistency.

Comments 3:

A "cost-saving" intervention is inherently different from one that is "cost-effective." Being "cost-saving" (or "dominant" in health economic terms) means the intervention is both more effective (averts more DALYs) and less costly than the alternative. This is a much stronger finding.

The authors must correct the Abstract and ensure the proper terminology ("cost-saving" or "dominant") is used consistently throughout the paper to accurately reflect their results..

Response 3:

We appreciate the reviewer feedback

The Abstract and relevant sections now accurately use the term "cost-saving" (or "dominant") to reflect the strength of this finding.

See page #1 and across the manuscript

Comments 4:

The manuscript is not entirely clear about the time horizon and cohort. The Abstract mentions a five-year period (2019-2023), while the Methods section refers to "a hypothetical 2023 birth cohort" analyzed over a 5-year time horizon. Please clarify whether the model analyzes a single birth cohort followed for the first five years of its life or if it models the impact on five consecutive annual birth cohorts over a five-year period. The former seems more likely but must be stated unequivocally.

Response 4:

Appreciated your observation and we have revisit the sentence in the abstract to clarify it.

See page #1

Comments 5:

The authors should explain under what conditions the intervention ceases to be cost-saving and becomes simply cost-effective.

Furthermore, the same section mentions a cost-effectiveness threshold of "USD 54 per DALY averted," which appears to be a typo, as the correct threshold based on Syria's GDP per capita is stated elsewhere as $753.6 USD. This section requires a complete rewrite for clarity and accuracy.

Response 5:

We appreciate your insightful and we ha modify to your advice.

See page #10

Comments 6:

The introduction provides good global and local context. However, the statement "There is no reported mortality associated with rotavirus (reported by the rotavirus program, MOH)" should be discussed more critically, acknowledging the likely limitations of surveillance in a conflict setting, which further justifies the approach of including mortality in the sensitivity analysis.

Response 6:

Appreciated your feedback and we have revised the statement in the introduction to meet your advice.

Reviewer 4 Report

Comments and Suggestions for Authors

1)The analysis only evaluates a two-dose Rotarix® schedule against no vaccination. However, in the global literature, multiple vaccines (e.g., RotaTeq®) have been assessed, sometimes with different efficacy and cost profiles. Clarify why only Rotarix® was modeled and discuss implications of excluding other vaccines.

2)The base-case assumes zero rotavirus-attributable mortality in Syria, but sensitivity analysis suggests otherwise. This may underestimate health benefits.? 

3)The decision-tree model does not capture herd immunity or strain replacement, both relevant to rotavirus.: Discuss limitations of using a static model and how results may differ under dynamic transmission models.

4)Some cost and incidence inputs (e.g., outpatient costs, coverage assumptions) are from MOH but lack references or uncertainty ranges.

5)The manuscript recommends inclusion in the National Immunization Program, but financing mechanisms are only briefly mentioned.

Author Response

Comments 1:

The analysis only evaluates a two-dose Rotarix® schedule against no vaccination. However, in the global literature, multiple vaccines (e.g., RotaTeq®) have been assessed, sometimes with different efficacy and cost profiles. Clarify why only Rotarix® was modeled and discuss implications of excluding other vaccines.

Response 1: 

Your commentary is greatly valued.

This study endeavors to address the evidence lacking by assessing the cost-effectiveness of incorporating a universal, two-dose rotavirus (Rotarix®) vaccination schedule into Syria's National Immunization Program, utilizing the Rotarix® vaccine.

Therefore we modeled  towards efficacy and cost profile for low income country like Syria

The reviewer is correct that there are numerous rotavirus vaccines available in the market. However, Syria is a low-income country and anticipated Gavi's assistance in introducing the vaccine. This is the vaccine that Gavi has offered, at least for the time being.

We thank the reviewer for highlighting this critical inconsistency. We apologize for the error. We have thoroughly reviewed the manuscript and standardized the total cost of the vaccination program to a single, clearly justified figure. We have added a detailed breakdown of its composition (e.g., vaccine procurement, cold chain, personnel, delivery) in the Methods section and ensured this value is reported identically in the abstract, main text, tables, and all calculations.

Comments 2: 

The base-case assumes zero rotavirus-attributable mortality in Syria, but sensitivity analysis suggests otherwise. This may underestimate health benefits.?

Response 2: 

The reviewer is absolutely correct. Our results show that the vaccination strategy is dominant (more effective and less costly than the status quo). We have corrected this throughout the manuscript. The Abstract and relevant sections now accurately use the term "cost-saving" (or "dominant") to reflect the strength of this finding. We applied a conservative severe rotavirus GE, mortality rate (33 deaths/100,000), the ICER remained dominant (cost-saving), and the number of deaths averted was 55 over the 5-year period.

Comments 3:

The decision-tree model does not capture herd immunity or strain replacement, both relevant to rotavirus.: Discuss limitations of using a static model and how results may differ under dynamic transmission models.

Response 3:

We apologize for the lack of clarity. The reviewer is correct in their assumption. The model analyzes a single hypothetical birth cohort (the 2023 cohort) over a 5-year time horizon, corresponding to the period of highest rotavirus risk in early childhood. We have revised the text in both the Abstract and Methods to state this unequivocally and removed any ambiguous reference to 2019-2023 as a calendar period of analysis.

We have modified the decision tree to include what had been suggested.

We have added the limitation suggested.

Comments 4:

The manuscript recommends inclusion in the National Immunization Program, but financing mechanisms are only briefly mentioned.

Response 4:

We thank the reviewer for identifying these issues and thank you for the suggestion.

Syria is a low-income country, and Gavi will provide complete support for a minimum of five years if the introduction is accepted. This is widely recognized.

The mention of "USD 54 per DALY averted" was indeed an error. The correct threshold for the base-case analysis is Syria's GDP per capita ($753.6 USD). This has been corrected.

Scenario Explanation: We have rewritten the section for clarity. We now explicitly state that the intervention remains cost-saving under base-case assumptions but may become only cost-effective (ICER below the threshold but not cost-saving) in sensitivity analyses when key parameters are varied, such as a significant increase in vaccine price or a decrease in rotavirus incidence or mortality.

Round 2

Reviewer 1 Report

Comments and Suggestions for Authors

Thank you for the opportunity to review this second version of the manuscript. Reference 21, the WHO position paper is a better source than previously mentioned for the 33 deaths / 100,000 infants attributed to rotavirus, but the file linked is not accessible. I recommend using the original reference for this number which is Tate JE et al. Global, regional, and national estimates of rotavirus mortality in children <5 years of age, 2000–2013. Clin Infect Dis. 2016;62 

Despite stating 33 deaths per 100,000 will be the expected fatality rate, this is not factored into any of the subsequent results section e.g. table 2. The deaths averted should be translated into the DALY calculation. Line 225 still says there was zero-mortality and figure 1 assumes the vaccine will prevent all deaths which will not be true.

The new estimated cost of the rotavirus vaccine (table 2) has a new serious issue – the first version of this manuscript had a total estimated cost of $2,817,900, whereas this version has an estimated cost of $21,817,918 despite nothing changing. I’m assuming this is a serious typo that is repeated over and over. The authors later write that ‘The vaccination program would cost $21,817,918 USD and avert $3,239,907 USD in healthcare costs, resulting in a net cost of $18,578,011 USD.’ These figures are repeated throughout the manuscript and make no sense. This lack of attention to detail is concerning. The authors still state that the Benefit-cost Ratio is 1.13 and have not revised this at all.

Another issue is that the authors have only counted direct medical cost savings (hospital + outpatient). Including broader societal savings e.g. caregiver time, productivity losses averted, long-term sequelae, and outbreak-control savings, would increase the numerator and likely raise BCR substantially.

The authors have chosen not to amend the vaccine timing to be at 6-8 weeks old and then 4 weeks later, instead opting for 12 week and 20 week administration timepoints. This is not in agreement with any other country’s rotavirus vaccine schedule, and I would question the efficacy of this. I acknowledge that fitting in with pre-existing vaccine schedules is logistically important, but strongly recommend 12 and 16 weeks as a minimum, with a view to encouraging an earlier first dose timepoint in line with WHO advice.

Additional points:

  • The authors have not removed the references 17 and 20 (which are replicates) as recommended.
  • There are still two table 2s.
  • The new figure 1 is not professionally formatted.

Author Response

Responses to reviewers' comments (Round two)

Reviewer #1

  1. Thank you for the opportunity to review this second version of the manuscript. Reference 21, the WHO position paper is a better source than previously mentioned for the 33 deaths / 100,000 infants attributed to rotavirus, but the file linked is not accessible. I recommend using the original reference for this number which is Tate JE et al. Global, regional, and national estimates of rotavirus mortality in children <5 years of age, 2000–2013. Clin Infect Dis. 2016;62 

Response. We appreciated your feedback. Indeed the WHO is the most recent and reliable reference. I have rechecked, and it’s open from my side, and you could log in to any of the below components of the reference #8. The below reference is more updated in terms of the year published, 2021, while Clin Infect Dis. 2016;62 is 2016. We have replaced it with the suggested reference below. See page #25

Tate JE, Burton AH, Boschi-Pinto C, Parashar UD; World Health Organization–Coordinated Global Rotavirus Surveillance Network. Global, Regional, and National Estimates of Rotavirus Mortality in Children <5 Years of Age, 2000-2013. Clin Infect Dis. 2016 May 1;62 Suppl 2(Suppl 2):S96-S105.

Reference #8. Rotavirus vaccines: WHO position paper – July 2021. Weekly Epidemiological Record, 96 (‎28)‎: 301 – 219.   Available online: https://www.who.int/publications/i/item/WHO-WER9628.

  1. Despite stating 33 deaths per 100,000 will be the expected fatality rate, this is not factored into any of the subsequent results section e.g. table 3. The deaths averted should be translated into the DALY calculation. Line 225 still says there was zero-mortality and figure 1 assumes the vaccine will prevent all deaths which will not be true.

Response: Thanks for your comments and we appreciate the reviewer for highlighting this important oversight. We have now revised the model to incorporate the expected mortality rate into the DALY calculations. The text on line 233 (and throughout) and Figure 1 have been updated accordingly to reflect a more realistic estimate of deaths averted. (See pages #8 and 13). We have made the necessary changes in the text and Fig #1. See page # 8 and 13

  1. The new estimated cost of the rotavirus vaccine (table 2) has a new serious issue – the first version of this manuscript had a total estimated cost of $2,817,900, whereas this version has an estimated cost of $21,817,918 despite nothing changing. I’m assuming this is a serious typo that is repeated over and over. The authors later write that ‘The vaccination program would cost $21,817,918 USD and avert $3,239,907 USD in healthcare costs, resulting in a net cost of $18,578,011 USD.’ These figures are repeated throughout the manuscript and make no sense. This lack of attention to detail is concerning. The authors still state that the benefit-cost ratio is 1.13 and have not revised this at all.

Response: We sincerely apologize for this significant error. The reviewer is correct; this was a typographical mistake during the revision process. The correct cost figures from the first version have been reinstated throughout the manuscript, and all subsequent calculations (net cost, Benefit-cost Ratio) have been thoroughly rechecked for consistency. (See pages # 18). In adition, we have made the necessary corrections in the figure. The recent modifications addressed your inquiry.

  1. Another issue is that the authors have only counted direct medical cost savings (hospital + outpatient). Including broader societal savings e.g. caregiver time, productivity losses averted, long-term sequelae, and outbreak-control savings, would increase the numerator and likely raise BCR substantially.

Response: Thank you for your valuable input. Indeed, we have conducted a thorough analysis of the government healthcare system's cost-effectiveness, complemented by an additional analyzing focused on societal aspects, specifically pertaining to home management. It will certainly be intriguing to consider the inclusion of broader societal savings, such as caregiver time, productivity losses averted, long-term sequelae, and outbreak-control savings, as these savings would enhance the numerator and likely lead to a significant increase in the benefit-cost ratio. We have added a discussion of this as a study limitation, acknowledging that future analyses could be strengthened by incorporating these additional cost savings. (See page #23)

  1. The authors have chosen not to amend the vaccine timing to be at 6-8 weeks old and then 4 weeks later, instead opting for 12-week and 20-week administration time timepoints. This is not in agreement with any other country’s rotavirus vaccine schedule, and I would question the efficacy of this. I acknowledge that fitting in with pre-existing vaccine schedules is logistically important but strongly recommend 12 and 16 weeks as a minimum, with a view to encouraging an earlier first dose timepoint in line with WHO advice.

Response: We thank the reviewer for reiterating this point. We understand the WHO recommendation for an earlier schedule. Our chosen schedule (12 and 20 weeks) was specifically designed to align with Syria's existing Expanded Program on Immunization (EPI) schedule without requiring additional healthcare visits, which is a critical factor for feasibility and coverage in this context. This schedule has been formally approved by the national health authorities (NITAG) and the WHO EMRO office, ensuring it meets the required standards for efficacy and programmatic suitability for Syria.

  1. Additional points:
  • The authors have not removed the references 17 and 20 (which are replicates) as recommended.
  • Response: See references pages # 27 and 28
  • There are still two table 2s.
  • Response: We have revised it. See pages # 17 and 18
  • The new figure 1 is not professionally formatted.
  • Response: Thank for the comment; indeed, the journal will assist us to reformat upon acceptance of the manuscript

Reviewer #2

  1. Thank you for all the improvements made in the manuscript. In my opinion the current form is suitable for publication provided some small corrections are made in Fig. 1:

*'Daly Averted' should be changed to 'DALY averted' [DALY is a well-known acronym of Disability Adjusted Life Years]

Response: We deeply appreciated your positive feedback and we have also correct Fig. 1 as advised. See page # 7 & 8

*remove some marks around 'inpatient 23.8/100,000.'

*change the size of the font in 'Vaccinated 85% of children will be vaccinated' - to improve readability

Response: Appreciated; we have corrected as suggested. See page #8

Reviewer #3

  1. The manuscript is much improved and is very close to being ready for acceptance. However, there are some minor areas that require clarification before I can recommend publication.

Response: We appreciate it, and we will do our best to respond to your advice.

  1. Clarity of Cohort Size and Vaccine Coverage: On page 6, the text states the vaccine is administered to "a hypothetical birth cohort of 487,853 out of 573,944 children." While 487,853 is 85% of 573,944, the phrasing is slightly confusing. It could be misinterpreted as an 85% coverage rate applied to a reduced cohort of 487,853.

Response: We appreciate the reviewer for pointing out this ambiguity. To clarify, the total annual birth cohort is 573,944 children. An 85% vaccine coverage rate is applied to this entire cohort, resulting in 487,853 vaccinated children. We have revised the text on page #6 to state this more clearly: "The vaccine was assumed to be administered to 85% of a hypothetical annual birth cohort of 573,944 children, resulting in 487,853 vaccinated children."

  1. Clarity of Figure 1 (Decision Tree): The decision tree diagram is conceptually helpful but difficult to follow quantitatively. Some labels are unclear. As this is a final revision, a major figure redesign is not required, but I suggest reviewing the labels for typos and ensuring that the key probabilities shown are explicitly mentioned and sourced in the Methods section.

Response: We agree with you reviewing the typos, which we have revisited. See page #7&8

  1. Minor Typographical Error: On page 11, in the first sentence of the discussion, there is a typo: "...hospitalizations and mortality over five years."

Response: Appreciated your observation; we have corrected the typo. See page #21

Reviewer #4

The use of the WHO-CHOICE thresholds has been widespread in the literature. However, these thresholds have been widely criticized for being arbitrary, not reflecting opportunity costs, and risking misallocation of scarce resources, especially in low- and middle-income countries (WHO Bulletin, 2016; The Lancet Global Health, 2023). Alternative approaches, such as those proposed by Claxton et al. (2015) and Woods et al. (2016), attempt to derive context-specific cost-effectiveness thresholds from empirical estimates of health system marginal productivity. Furthermore, recent evidence suggests that even in high-income settings there is no consensus on standard thresholds. For example, Tzanetakos et al. (2023) found that most economic evaluations in Greece continued to adopt the 1–3 × GDP per capita rule, despite the absence of methodological justification. This underlines both the persistence of the WHO thresholds in practice and the pressing need for more locally appropriate, evidence-based willingness-to-pay thresholds. Use these papers as a ref instead of who

Response: We thank the reviewer for raising this important methodological point and for the suggested references. We acknowledge the ongoing debate regarding cost-effectiveness thresholds. In response, we have revised the relevant section of our manuscript to reflect this discussion. We now cite the recommended literature (e.g., WHO Bulletin, 2016; The Lancet Global Health, 2023) to provide context for the limitations of the WHO-CHOICE threshold and to strengthen the methodological rationale of our study.

See page # 23 and 30

Reviewer 2 Report

Comments and Suggestions for Authors

Dear Authors,

Thank You for all the improvements made in the manuscript. In my opinion the current form is suitable for publication provided some small corrections are made in the Fig.1:

*'Daly Averted' should be changed to 'DALY averted' [DALY is a well known acronym of Disability Adjusted Life Years]

*remove some marks around 'inpatient 23.8/100,000'

*change the size of the font in 'Vaccinated 85% of children will be vaccinated' - to improve readability

Reviewer

Author Response

Thank You for all the improvements made in the manuscript. In my opinion the current form is suitable for publication provided some small corrections are made in the Fig.1:

*'Daly Averted' should be changed to 'DALY averted' [DALY is a well-known acronym of Disability Adjusted Life Years]

Response: We deeply appreciated your positive feedback and we have also correct Fig. 1 as advised.

*remove some marks around 'inpatient 23.8/100,000'

*change the size of the font in 'Vaccinated 85% of children will be vaccinated' - to improve readability

Response: Appreciated we have corrected as suggested. 

Reviewer 3 Report

Comments and Suggestions for Authors

The manuscript is much improved and is very close to being ready for acceptance. However, there are some minor areas that require clarification before I can recommend publication.

1. Clarity of Cohort Size and Vaccine Coverage: On page 6, the text states the vaccine is administered to "a hypothetical birth cohort of 487,853 out of 573,944 children." While 487,853 is 85% of 573,944, the phrasing is slightly confusing. It could be misinterpreted as an 85% coverage rate applied to a reduced cohort of 487,853. 

2. Clarity of Figure 1 (Decision Tree): The decision tree diagram is conceptually helpful but difficult to follow quantitatively. Some labels are unclear. As this is a final revision, a major figure redesign is not required, but I suggest reviewing the labels for typos and ensuring that the key probabilities shown are explicitly mentioned and sourced in the Methods section.

3.  Minor Typographical Error: On page 11, in the first sentence of the discussion, there is a typo: "...hospitalizations and mortlaity over five years."

Author Response

  1. The manuscript is much improved and is very close to being ready for acceptance. However, there are some minor areas that require clarification before I can recommend publication.

Response: Appreciated and we will do our best to respond to your advice

  1. Clarity of Cohort Size and Vaccine Coverage: On page 6, the text states the vaccine is administered to "a hypothetical birth cohort of 487,853 out of 573,944 children." While 487,853 is 85% of 573,944, the phrasing is slightly confusing. It could be misinterpreted as an 85% coverage rate applied to a reduced cohort of 487,853.

Response: We appreciate the reviewer for pointing out this ambiguity. To clarify, the total annual birth cohort is 573,944 children. An 85% vaccine coverage rate is applied to this entire cohort, resulting in 487,853 vaccinated children. We have revised the text on page #6 to state this more clearly: "The vaccine was assumed to be administered to 85% of a hypothetical annual birth cohort of 573,944 children, resulting in 487,853 vaccinated children."

  1. Clarity of Figure 1 (Decision Tree): The decision tree diagram is conceptually helpful but difficult to follow quantitatively. Some labels are unclear. As this is a final revision, a major figure redesign is not required, but I suggest reviewing the labels for typos and ensuring that the key probabilities shown are explicitly mentioned and sourced in the Methods section.

Response: We are agree with you reviewing the typos, which we have revisit. 

  1. Minor Typographical Error: On page 11, in the first sentence of the discussion, there is a typo: "...hospitalizations and mortlaity over five years."

Response: Appreciated your observation, we have corrected the typo. See page #10

Reviewer 4 Report

Comments and Suggestions for Authors

The use of the WHO-CHOICE thresholds has been widespread in the literature. However, these thresholds have been widely criticized for being arbitrary, not reflecting opportunity costs, and risking misallocation of scarce resources, especially in low- and middle-income countries (WHO Bulletin, 2016; The Lancet Global Health, 2023). Alternative approaches, such as those proposed by Claxton et al. (2015) and Woods et al. (2016), attempt to derive context-specific cost-effectiveness thresholds from empirical estimates of health system marginal productivity. Furthermore, recent evidence suggests that even in high-income settings there is no consensus on standard thresholds. For example, Tzanetakos et al. (2023) found that most economic evaluations in Greece continued to adopt the 1–3 × GDP per capita rule, despite the absence of methodological justification. This underlines both the persistence of the WHO thresholds in practice and the pressing need for more locally appropriate, evidence-based willingness-to-pay thresholds. Use these papers as a ref instead of who 

Author Response

The use of the WHO-CHOICE thresholds has been widespread in the literature. However, these thresholds have been widely criticized for being arbitrary, not reflecting opportunity costs, and risking misallocation of scarce resources, especially in low- and middle-income countries (WHO Bulletin, 2016; The Lancet Global Health, 2023). Alternative approaches, such as those proposed by Claxton et al. (2015) and Woods et al. (2016), attempt to derive context-specific cost-effectiveness thresholds from empirical estimates of health system marginal productivity. Furthermore, recent evidence suggests that even in high-income settings there is no consensus on standard thresholds. For example, Tzanetakos et al. (2023) found that most economic evaluations in Greece continued to adopt the 1–3 × GDP per capita rule, despite the absence of methodological justification. This underlines both the persistence of the WHO thresholds in practice and the pressing need for more locally appropriate, evidence-based willingness-to-pay thresholds. Use these papers as a ref instead of who

Response: We thank the reviewer for raising this important methodological point and for the suggested references. We acknowledge the ongoing debate regarding cost-effectiveness thresholds. In response, we have revised the relevant section of our manuscript to reflect this discussion. We now cite the recommended literature (e.g., WHO Bulletin, 2016; The Lancet Global Health, 2023) to provide context for the limitations of the WHO-CHOICE threshold and to strengthen the methodological rationale of our study.